# GENERALIZING FEW-SHOT NAS WITH GRADIENT MATCHING

**Shoukang Hu[1]\* Ruochen Wang[2]\* Lanqing Hong[3] Zhenguo Li[3] Cho-Jui Hsieh[2] Jiashi Feng[4]**
[1]The Chinese University of Hong Kong    [2]University of California, Los Angeles
[3]Huawei Noah's Ark Lab    [4]National University of Singapore
skhu@se.cuhk.edu.hk    ruocwang@ucla.edu    {honglanqing, Li.Zhenguo}@huawei.com
chohsieh@cs.ucla.edu    jshfeng@gmail.com

## ABSTRACT

Efficient performance estimation of architectures drawn from large search spaces is essential to Neural Architecture Search. One-Shot methods tackle this challenge by training one supernet to approximate the performance of every architecture in the search space via weight-sharing, thereby drastically reducing the search cost. However, due to coupled optimization between child architectures caused by weight-sharing, One-Shot supernet's performance estimation could be inaccurate, leading to degraded search outcomes. To address this issue, Few-Shot NAS reduces the level of weight-sharing by splitting the One-Shot supernet into multiple separated sub-supernets via edge-wise (layer-wise) exhaustive partitioning. Since each partition of the supernet is not equally important, it necessitates the design of a more effective splitting criterion. In this work, we propose a gradient matching score (GM) that leverages gradient information at the shared weight for making informed splitting decisions. Intuitively, gradients from different child models can be used to identify whether they agree on how to update the shared modules, and subsequently to decide if they should share the same weight. Compared with exhaustive partitioning, the proposed criterion significantly reduces the branching factor per edge. This allows us to split more edges (layers) for a given budget, resulting in substantially improved performance as NAS search spaces usually include dozens of edges (layers). Extensive empirical evaluations of the proposed method on a wide range of search spaces (NASBench-201, DARTS, MobileNet Space), datasets (cifar10, cifar100, ImageNet) and search algorithms (DARTS, SNAS, RSPS, ProxylessNAS, OFA) demonstrate that it significantly outperforms its Few-Shot counterparts while surpassing previous comparable methods in terms of the accuracy of derived architectures. Our code is available at https://github.com/skhu101/GM-NAS.

## 1 INTRODUCTION

In recent years, there has been a surge of interest in Neural Architecture Search (NAS) (Stanley & Miikkulainen, 2002; Zoph & Le, 2017; Pham et al., 2018; Real et al., 2019; Liu et al., 2019) for its ability to identify high-performing architectures in a series of machine learning tasks. Pioneering works in this field require training and evaluating thousands of architectures from scratch, which consume huge amounts of computational resources (Miikkulainen et al., 2019; Zoph & Le, 2017; Zoph et al., 2018). To improve the search efficiency, One-Shot NAS (Pham et al., 2018; Liu et al., 2019; Bender et al., 2018) proposes to train a single weight-sharing supernet (Shot) that encodes every architecture in the search space as a sub-path, and subsequently uses this supernet to estimate the performance of the underlying architectures efficiently. The supernet is represented as a directed acyclic graph (DAG), where each edge is associated with a set of operations. In the One-Shot supernet, child models share weight when their paths in the DAG overlap. This way, One-Shot methods manage to cut the search cost down to training a single supernet model while still achieving state-of-the-art performances.

---

*\*Equal Contribution*

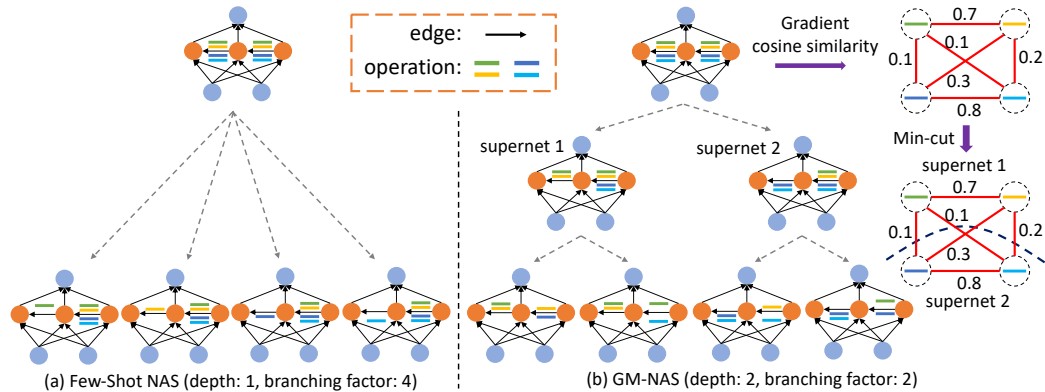

Figure 1: An illustration of the supernet partitioning schema in Few-Shot NAS v.s. GM-NAS (ours)

Despite the search efficiency, training the weight-sharing supernet also induces coupled optimization among child models. Consequently, the supernet suffers from degenerated search outcomes due to inaccurate performance estimation, especially on top architectures (Bender et al., 2018; Yu et al., 2020b; Pourchot et al., 2020; Zhang et al., 2020b; Zhao et al., 2021b). To reduce the level of weight-sharing, Few-Shot NAS (Zhao et al., 2021b) proposes to split the One-Shot supernet into multiple independent sub-supernets via edge-wise exhaustive partitioning. Concretely, it assigns every operation on the selected edges to a separated sub-supernet, with weight-sharing enabled only within each sub-supernet (Figure 1 (a)). This way, architectures containing different operations on the split edge are divided into different sub-supernets, thereby disabling weight-sharing between them. Although the performance of Few-Shot NAS surpasses its One-Shot counterpart, its splitting schema - naively assigning each operation on an edge to a separated sub-supernet - is not ideal. Some operations might behave similarly to each other and thus can be grouped into the same sub-supernet with little harm. In such cases, dividing them up could be a waste of precious search budgets while bringing little benefit to the performance. On the other hand, the gain from dividing dissimilar operations considerably out-weights that of dividing similar ones.

The above analysis necessitates the design of an effective splitting criterion to distinguish between these two cases. Consider two child models that contain the same operations on all but the to-be-split edge of the supernet. Intuitively, they should not use one copy of weight if their training dynamics at the shared modules are dissimilar. Concretely, when these two networks produce mismatched gradients for the shared weight, updating the shared module under them would lead to a zigzag SGD trajectory. As a result, the performance estimation of these networks might not reflect their true strength. This can be further motivated by viewing the supernet training as multi-criteria optimization (Fliege & Svaiter, 2000; Brown & Smith, 2005), where each criterion governs the optimization of one child model; Only similar update directions from different objectives could reduce the loss of all child networks. Otherwise, the performance of some child networks (objectives) would deteriorate. Based on these inspirations, we propose to directly use the gradient matching score (GM) as the supernet splitting criterion. The splitting decision can be made via graph clustering over the sub-supernets, with graph links weighted by the GM scores (Figure 1 (b)). Utilizing the proposed splitting schema, we generalize the supernet splitting of Few-Shot NAS to support arbitrary branching factors (number of children at each node of the partition tree). With a much lower branching factor, we could afford to split three times more edges compared with Few-Shot NAS under the same search budget, and achieve superior search performance.

We conduct extensive experiments on multiple search spaces, datasets, and base methods to demonstrate the effectiveness of the proposed method, codenamed GM-NAS. Despite its simplicity, GM-NAS consistently outperforms its One-Shot and Few-Shot counterparts. On DARTS Space, we achieve a test error of 2.34%, ranked top among SOTA methods. On the MobileNet Space, GM-NAS reaches 19.7% Top-1 test error on ImageNet, surpassing previous comparable methods.

## 2 RELATED WORK

One-Shot NAS with Weight-Sharing aims at addressing the high computational cost of early NAS algorithms (Bender et al., 2018; Liu et al., 2019; Li & Talwalkar, 2020). Concretely, One-Shot NAS builds one single supernet that includes all child models in the search space as sub-paths, and allows the child models to share weight when their paths overlap. Architecture search can be conducted by

training the supernet once and using it as the performance estimator to derive the best child model from the supernet. As a result, One-Shot NAS reduces the search cost down to training one single model (the supernet).

Despite the search efficiency, Weight-sharing technique adopted by One-Shot NAS also posts a wide range of inductive biases due to coupled optimization, such as operation co-adaptation (Bender et al., 2018; Li et al., 2019a), poor generalization ability (Zela et al., 2020; Chen et al., 2021), and distorted ranking correlations especially among top architectures (Zhang et al., 2020b; Bender et al., 2018; Zhao et al., 2021b). As a result, the performance estimation of child models from the supernet could be inaccurate, and thus degrades the search results.

Several lines of methods have been proposed to address the degenerated search performance caused by weight-sharing. Search space pruning methods identify and progressively discard poor-performing regions of the search space, so that other models do not need to share weight with candidates from these regions (Liu et al., 2018; Li et al., 2019a; Chen et al., 2021; Li et al., 2020a; Hu et al., 2020b). Distribution learning methods aim at inferring a sampling distribution that biases towards top performers (Xie et al., 2019; Chen et al., 2021; Hu et al., 2020a; Dong & Yang, 2019). Recently, there emerges a new orthogonal line of work that directly reduces the level of weight-sharing by partitioning the search space into multiple sub-regions, with weight-sharing enabled only among models inside each sub-regions (Zhang et al., 2020b; Zhao et al., 2021b). Zhang et al. (2020b) shows on a reduced search space that this treatment improves the ranking correlation among child models. Zhao et al. (2021b) further proposes Few-Shot NAS: an edge-wise exhaustive partitioning schema that splits the One-Shot supernet into multiple sub-supernets, and obtains significantly improved performance over One-Shot baselines. Our work generalizes Few-Shot NAS to arbitrary branching factors by utilizing gradient matching score as the splitting criteria and formulating the splitting as a graph clustering problem.

## 3 METHOD

### 3.1 FROM ONE-SHOT NAS TO FEW-SHOT NAS

**One-Shot NAS** One-Shot NAS represents the search space as a directed acyclic graph (DAG), where each node denotes a latent feature map and each edge ($e$) contains operations $o$ from the set $\mathcal{O}^{(e)}$. This way, every child model in the search space can be represented as one path in the DAG (a.k.a. supernet). The search process is conducted by first optimizing the supernet once and subsequently using it as the performance estimator to derive the best architectures. One-Shot NAS induces the maximum level of weight-sharing: a child model shares weights with every other model in the search space as long as they include the same operation(s) on some edge(s). Although weight-sharing supernet drastically reduces the search cost, it also leads to inaccurate performance estimation of child models, as pointed out by several previous works (Zela et al., 2020; Bender et al., 2018; Zhang et al., 2020b; Zhao et al., 2021b; Wang et al., 2021b).

**Few-Shot NAS** To address the aforementioned issue in One-Shot NAS, Zhao et al. (2021b) proposes Few-Shot NAS that leverages supernet splitting to reduce the level of weight-sharing in One-Shot supernet. Few-Shot NAS divides the One-Shot supernet into multiple sub-supernets (each corresponding to a "shot"), where weight-sharing occurs only among the child models that belong to the same sub-supernet. Concretely, it adopts an edge-wise splitting schema: It first randomly selects a target compound edge from the supernet and then assigns each operation on the target edge into a separated sub-supernet, while keeping the remaining edges unchanged. As a result, child models containing different operations on the target edge are assigned to different sub-supernets, and therefore do not share weight with each other. As shown in Figure 1 (a), this supernet splitting schema forms a partition tree of supernets, with a branching factor $B$ equals to the number of operations on the edge ($|\mathcal{O}^{(e)}|$).

### 3.2 GENERALIZED SUPERNET SPLITTING WITH ARBITRARY BRANCHING FACTORS

Due to the exhaustive splitting schema, Few-Shot NAS suffers from a high branching factor per split. Suppose we perform splits on $T$ edges, the number of leaf-node sub-supernets becomes $|\mathcal{O}^{(e)}|^T$. On DARTS Space, where the supernet contains 7 operations per edge and 14 edges per cell, splitting

merely two edges lead to $7^2 = 49$ sub-supernets. Consequently, conducting architecture search over these many sub-supernets induces prohibitively large computational overhead. For this reason, Few-Shot NAS could only afford to split very few edges (in fact, one single split in most of its experiments). This could be suboptimal for many popular NAS search spaces as the supernets usually contain multiple edges/layers (14 for DARTS Space and 22 for MobileNet Space), and the decision on a single edge might not contribute much to the performance.

Given a fixed search budget, measured by the total number of leaf-node sub-supernets, there exists a trade-off between the branching factor and the total number of splits. Exhaustive splitting can be viewed as an extreme case with a maximum branching factor. However, naively separating all operations on an edge might be unnecessary: Since some operations might behave similarly, splitting them into separated sub-supernets wastes a lot of search budgets while enjoying little benefit. In such a case, we could group these operations into the same sub-supernet with minor sacrifice. This reduces the branching factor, allowing us to split more edges for a predefined budget and improve the search performance on large NAS search spaces with many edges/layers. We term this splitting schema as *Generalized Supernet Splitting with arbitrary branching factors*.

We provide a formal formulation for this proposed splitting schema. At each split, we select an edge $e$ and divide the operations on it into $B$ disjoint partitions: $\mathcal{O}^{(e)} = \bigcup_{b_e=1\cdots B} \mathcal{O}_{b_e}^{(e)}$. $B$ is thus the branching factor. For Few-Shot NAS, $B = |\mathcal{O}^{(e)}|$, and $\mathcal{O}_{b_e}^{(e)}$ is simply a unit set with one operation from edge $e$. When $B < |\mathcal{O}^{(e)}|$, more than one operations on the target edge will be assigned to the same sub-supernet, i.e. $|\mathcal{O}_{b_e}^{(e)}| > 1$. Let $\mathcal{E}$ denote the set of all edges and $\mathcal{E}_t$ be the set of partitioned edges after $t$ splits, then any sub-supernet generated after the $t$-th split can be uniquely determined by the partitioning set $\mathcal{P}_t = \{(e, \mathcal{O}_{b_e}^{(e)}) | e \in \mathcal{E}_t\}$, where $\mathcal{P}_t$ contains tuples $(e, \mathcal{O}_{b_e}^{(e)})$ that record which operations on edge $e$ get assigned to this sub-supernet. Let $A_{\mathcal{P}_t}$ denote a sub-supernet with partition set $\mathcal{P}_t$, then

$$A_{\mathcal{P}_t} = \left( \bigcup_{(e, \mathcal{O}_{b_e}^{(e)}) \in \mathcal{P}_t} \mathcal{O}_{b_e}^{(e)} \right) \bigcup \left( \bigcup_{e \in \mathcal{E} \setminus \mathcal{E}_t} \mathcal{O}^{(e)} \right). \tag{1}$$

### 3.3 Supernet Splitting via Graph Min-cut with Gradient Matching Score

Generalized Supernet Splitting necessitates the design of a splitting criterion for grouping operations, which leads to a question as to how to decide which networks should or should not share weight. A naive strategy is to perform the random partition, with the underlying assumption that the weight-sharing between all child models are equally harmful. However, empirically we find that this treatment merely matches exhaustive splitting in terms of their performance, and in some cases even worsens the results. To demonstrate this, we compare random partition with the Few-Shot NAS (exhaustive) on NASBench201 (Dong & Yang, 2020) with four operations (skip, conv_1x1, conv_3x3, avgpool_3x3)[1]. For random partitioning, we split

Table 1: Comparison of different splitting schema on NASBench-201 and CIFAR-10. We run each method with four random seeds and report the mean accuracy of derived architectures. With the same amount of supernets, the search performance of random split with smaller branching factor is worse than Few-Shot NAS' exhaustive splitting. However, replacing random split with the proposed Gradient Matching criterion significantly improves the results (More on this later).

| Base | Split Criterion | Branch Factor | #Splits | #Supernets | Accuracy |
|------|----------------|---------------|---------|------------|----------|
| DARTS | Exhaustive | 4 | 1 | | 88.55% |
| | Random | 2 | 2 | 4 | 70.47% |
| | Gradient (ours) | 2 | 2 | | **93.95%** |
| RSPS | Exhaustive | 4 | 1 | | 88.96% |
| | Random | 2 | 2 | 4 | 88.83% |
| | Gradient (ours) | 2 | 2 | | **92.52%** |

two edges on its supernet ($T = 2$), and divide the operations on each edge into two groups randomly ($B = 2$). This leads to four sub-supernets, same as Few-Shot NAS with one single split. As shown in Table 1, random split degrades the search performance for both continuous and sampling-based One-Shot NAS. The comparison result reveals that the weight-sharing between some child models are much more harmful than others, and therefore need to be carefully distinguished.

If not all weight-sharing are equally harmful, how should we decide whether two models should be split into separated sub-supernets? Intuitively, networks should not share weight if their training dynamics mismatch at the shared modules. In other words, they disagree on how to update the

---

[1] Using 4 operations allow us to match the number of supernets between random and exhaustive split.

shared weight. In this case, the angles between gradient directions produced by two networks at the shared modules might be large, leading to a zigzag pattern of SGD trajectory. As a result, the performance estimation of these models in the supernet could be inaccurate, thereby degrading the performance of downstream architecture search task.

To show this, we evaluate the performance of a single model $A$ when updated together with another network $A_{sim}$ that produces similar gradients at shared weight, and compare the performance with the same model updated together with a network $A_{dissim}$ that produces dissimilar gradients. We generate $(A, A_{sim}, A_{dissim})$ by sampling from the NASBench-201 search space, in a way that they have the same operation (and thus share weights) on all but one edges (More on this in Appendix B.2). We then proceed to train $(A, A_{sim})$ together and similarly $(A, A_{dissim})$ together via Random Sampling with Parameter Sharing (RSPS) (Li & Talwalkar, 2020), and record $A$'s performance under these two cases. As shown in Table 2, $A$ achieves a much lower loss when updated together with $A_{sim}$, where gradients from two child models are similar at the shared module. The result indicates that weight-sharing could be more harmful between networks with dissimilar training dynamics.

Inspired by the above analysis, we propose to measure the harmfulness of weight-sharing between child models (hence sub-supernets) directly via gradient matching (GM). Concretely, consider two operations from an edge; if these operations, when enabled separately, lead to drastically different gradients on the shared edges, we give them higher priority for being split into different sub-supernets during the su-

Table 2: Performance of a network when updated with a model with similar training dynamics v.s. with a model with dissimilar training dynamics at their shared weight. The network achieves a much lower loss in the first case.

| Weight Sharing | Grad Similarity | Train Loss ($A_1$) | Valid Loss ($A_1$) |
|---|---|---|---|
| $(A, A_{sim})$ | $0.76 \pm 0.17$ | $\mathbf{0.74 \pm 0.18}$ | $\mathbf{0.86 \pm 0.10}$ |
| $(A, A_{dissim})$ | $0.12 \pm 0.06$ | $0.82 \pm 0.03$ | $0.99 \pm 0.03$ |

pernet partitioning. The entire splitting schema can be formulated as a graph clustering problem: Given a (sub-)supernet $\mathcal{A}_{\mathcal{P}_{t-1}}$, we evaluate the gradient of the supernet when each operation $o$ is enabled on edge $e_t$ separately, and then compute the cosine similarity between every pair of these gradients:

$$GM(A_{\mathcal{P}_{t-1}}|_{e_t=o}, A_{\mathcal{P}_{t-1}}|_{e_t=o'}) = COS\big[\nabla_{w_s}\mathcal{L}(A_{\mathcal{P}_{t-1}}|_{e_t=o}; w_s), \nabla_{w_s}\mathcal{L}(A_{\mathcal{P}_{t-1}}|_{e_t=o'}; w_s)\big] \quad (2)$$

where $A_{\mathcal{P}_{t-1}}|_{e_t=o}$ means to enable operation $o$ on edge $e_t$ of supernet $A_{\mathcal{P}_{t-1}}$, and $w_s$ is the shared weight. This leads to a densely connected graph (Figure 1 (b)) where the vertices are operations on $e_t$ and links between them are weighted by GM score computed in Eqn. (2). Therefore, supernet partitioning can be conducted by performing graph clustering on this graph, with the number of clusters equal to the desired branching factor. There exist many applicable algorithms for solving this problem, but $|\mathcal{O}^{(1)}|$ is usually small, we perform graph min-cut via brute-force search to divide the operations (supernets) into $B$ balanced groups. For $B = 2$, it can be written as:

$$\mathcal{U} = \arg\min_{\mathcal{U} \subseteq \mathcal{O}^{(e)}} \sum_{o \in \mathcal{U}, o' \in \mathcal{O}^{(e)} \setminus \mathcal{U}} GM(A_{\mathcal{P}_{t-1}}|_{e_t=o}, A_{\mathcal{P}_{t-1}}|_{e_t=o'}), \quad (3)$$

$$\text{s.t.} \ \lfloor |\mathcal{O}^{(e)}|/2 \rfloor \leq |\mathcal{U}| \leq \lceil |\mathcal{O}^{(e)}|/2 \rceil.$$

where $\{\mathcal{U}, \mathcal{O}^{(e)} \setminus \mathcal{U}\}$ are the obtained partitions. The proposed splitting schema substantially improves the search performance over exhaustive split and random split, as evidenced by Table 1.

## 3.4 THE COMPLETE ALGORITHM

**Edge selection using gradient matching score** Apart from smartly selecting which operations should be grouped within an edge, we can also use the gradient matching score to select which edge to split next. More specifically, the graph min-cut algorithm produces a cut cost for each edge - the sum of gradient matching scores of the cut links on the gradient similarity graph. This can serve as an edge importance measure for determining which edge to split on next, as a lower cut cost indicates that splitting on this edge first might relieve the adverse effect of weight-sharing to a larger extend. Empirically, we find that this edge scoring measure reduces the variance over random selection used in Few-Shot NAS, and also improves the performance (Section 5.1).

**Supernet splitting with restart** During the supernet splitting phase, we warmup the supernet's weight ($w$) for a few epochs before each split, in order to collect more accurate gradient information

along the optimization trajectory. Since the effect of weight-sharing has already kicked in during this phase, we re-initialize all the leaf-node sub-supernets after the final split is completed, and conduct architecture search over them from scratch. The complete algorithm is summarized in Algorithm 1 in the Appendix.

## 4 EXPERIMENTS

In this section, we conduct extensive empirical evaluations of the proposed method across various base methods (DARTS, RSPS, SNAS, ProxylessNAS, OFA) and search spaces (NASBench-201, DARTS, and MobileNet Space). Experimental results demonstrate that the proposed GM-NAS consistently outperforms its Few-Shot and One-Shot counterparts.

### 4.1 NASBENCH-201

We benchmark the proposed method on the full NASBench-201 Space (Dong & Yang, 2020) with five operations (none, skip, conv_1x1, conv_3x3, avgpool_3x3). We run the search phase of each method under four random seeds (0-3) and reports the mean accuracy of their derived architectures, as well as the standard deviation to capture the variance of the search algorithms. For our method, we split the operations on each edge into two groups (one group with three operations, another group with two operations), and cut two edges in total, amounting to four sub-supernets. Note that this is one supernet less than Few-Shot NAS, which uses five sub-supernets due to its exhaustive splitting schema.

Still, the proposed method improves over Few-Shot NAS by a large margin. As shown in Table 3, the architectures derived from GM DARTS achieve an average accuracy of 93.72% on CIFAR-10, leading to an improvement of 5.17% over Few-Shot DARTS and 39.24% over DARTS. To further test the generality of the proposed method over various One-Shot NAS algorithms, we also compare GM-NAS with Few-Shot NAS on sampling-based methods such as SNAS and RSPS. As before, our method consistently out-performs Few-Shot NAS and the One-Shot base methods by a substantial margin. Notably, when combined with the proposed method, SNAS matches the previous SOTA result (DrNAS) on CIFAR-10 and CIFAR-100.

Table 3: Comparison with state-of-the-art NAS methods on NASBench-201.

| Method | CIFAR-10 | | CIFAR-100 | | ImageNet-16-120 | |
|---|---|---|---|---|---|---|
| | validation | test | validation | test | validation | test |
| ResNet (He et al., 2016) | 90.83 | 93.97 | 70.42 | 70.86 | 44.53 | 43.63 |
| Random (baseline) | $90.93 \pm 0.36$ | $93.70 \pm 0.36$ | $70.60 \pm 1.37$ | $70.65 \pm 1.38$ | $42.92 \pm 2.00$ | $42.96 \pm 2.15$ |
| Reinforce (Zoph et al., 2018) | $91.09 \pm 0.37$ | $93.85 \pm 0.37$ | $70.05 \pm 1.67$ | $70.17 \pm 1.61$ | $43.04 \pm 2.18$ | $43.16 \pm 2.28$ |
| ENAS (Pham et al., 2018) | $39.77 \pm 0.00$ | $54.30 \pm 0.00$ | $10.23 \pm 0.12$ | $10.62 \pm 0.27$ | $16.43 \pm 0.00$ | $16.32 \pm 0.00$ |
| GDAS (Dong & Yang, 2019) | $90.01 \pm 0.46$ | $93.23 \pm 0.23$ | $24.05 \pm 8.12$ | $24.20 \pm 8.08$ | $40.66 \pm 0.00$ | $41.02 \pm 0.00$ |
| DSNAS (Hu et al., 2020a) | $89.66 \pm 0.29$ | $93.08 \pm 0.13$ | $30.87 \pm 16.40$ | $31.01 \pm 16.38$ | $40.61 \pm 0.09$ | $41.07 \pm 0.09$ |
| PC-DARTS (Xu et al., 2020) | $89.96 \pm 0.15$ | $93.41 \pm 0.30$ | $67.12 \pm 0.39$ | $67.48 \pm 0.89$ | $40.83 \pm 0.08$ | $41.31 \pm 0.22$ |
| DrNAS (Chen et al., 2021) | $\mathbf{91.55 \pm 0.00}$ | $\mathbf{94.36 \pm 0.00}$ | $\mathbf{73.49 \pm 0.00}$ | $\mathbf{73.51 \pm 0.00}$ | $\mathbf{46.37 \pm 0.00}$ | $46.34 \pm 0.00$ |
| RSPS (Li & Talwalkar, 2020) | $84.16 \pm 1.69$ | $87.66 \pm 1.69$ | $45.78 \pm 6.33$ | $46.60 \pm 6.57$ | $31.09 \pm 5.65$ | $30.78 \pm 6.12$ |
| Few Shot + RSPS | $85.40 \pm 1.28$ | $89.11 \pm 1.37$ | $58.59 \pm 3.45$ | $58.69 \pm 3.75$ | $34.24 \pm 1.45$ | $33.85 \pm 2.33$ |
| GM + RSPS | $\mathbf{89.09 \pm 0.40}$ | $\mathbf{92.70 \pm 0.53}$ | $\mathbf{68.36 \pm 0.91}$ | $\mathbf{68.81 \pm 1.28}$ | $\mathbf{42.65 \pm 1.04}$ | $\mathbf{43.47 \pm 1.02}$ |
| DARTS (Liu et al., 2019) | $39.77 \pm 0.00$ | $54.30 \pm 0.00$ | $38.57 \pm 0.00$ | $38.97 \pm 0.00$ | $18.87 \pm 0.00$ | $18.41 \pm 0.00$ |
| Few Shot + DARTS | $84.70 \pm 0.44$ | $88.55 \pm 0.02$ | $70.17 \pm 2.66$ | $70.16 \pm 2.87$ | $31.16 \pm 3.93$ | $30.09 \pm 4.43$ |
| GM + DARTS | $\mathbf{91.03 \pm 0.24}$ | $\mathbf{93.72 \pm 0.12}$ | $\mathbf{71.61 \pm 0.62}$ | $\mathbf{71.83 \pm 0.97}$ | $\mathbf{42.19 \pm 0.00}$ | $\mathbf{42.60 \pm 0.00}$ |
| SNAS (Xie et al., 2019) | $90.10 \pm 1.04$ | $92.77 \pm 0.83$ | $69.69 \pm 2.39$ | $69.34 \pm 1.98$ | $42.84 \pm 1.79$ | $43.16 \pm 2.64$ |
| Few Shot + SNAS | $90.47 \pm 0.48$ | $93.88 \pm 0.25$ | $71.28 \pm 1.29$ | $71.49 \pm 1.41$ | $46.17 \pm 0.35$ | $\mathbf{46.43 \pm 0.19}$ |
| GM + SNAS | $\mathbf{91.55 \pm 0.00}$ | $\mathbf{94.36 \pm 0.00}$ | $\mathbf{73.49 \pm 0.00}$ | $\mathbf{73.51 \pm 0.00}$ | $\mathbf{46.37 \pm 0.00}$ | $46.34 \pm 0.00$ |
| **optimal** | 91.61 | 94.37 | 73.49 | 73.51 | 46.77 | 47.31 |

### 4.2 DARTS SPACE

We further investigate the performance of the proposed method on the DARTS search space. To encourage fair comparisons with prior arts, we follow the same search and retrain settings as the original DARTS (Liu et al., 2019). Similar to the experiments on NASBench-201, we also run the **search phase** of each method under four random seeds (0-3) and report the best and average accuracy of **all derived architectures**, as well as the error bar to capture the variance of the search

Table 4: Comparison with state-of-the-art NAS methods on CIFAR-10.

| Architecture | Test Error(%) | | Param (M) | Search Cost (GPU Days) | Search Method |
|---|---|---|---|---|---|
| | Best | Avg | | | |
| DenseNet-BC (Huang et al., 2017) | 3.46 | - | 25.6 | - | manual |
| NASNet-A (Zoph et al., 2018) | 2.65 | - | 3.3 | 2000 | RL |
| AmoebaNet-A (Real et al., 2019) | - | $3.34 \pm 0.06$ | 3.2 | 3150 | evolution |
| AmoebaNet-B (Real et al., 2019) | - | $2.55 \pm 0.05$ | 2.8 | 3150 | evolution |
| PNAS (Liu et al., 2018) | - | $3.41 \pm 0.09$ | 3.2 | 225 | SMBO |
| ENAS (Pham et al., 2018) | 2.89 | - | 4.6 | 0.5 | RL |
| NAONet (Luo et al., 2018) | 3.53 | - | 3.1 | 0.4 | NAO |
| GDAS (Dong & Yang, 2019) | 2.93 | - | 3.4 | 0.3 | gradient |
| BayesNAS (Zhou et al., 2019) | $2.81 \pm 0.04$ | - | 3.4 | 0.2 | gradient |
| ProxylessNAS (Cai et al., 2019)[†] | 2.08 | - | 5.7 | 4.0 | gradient |
| PARSEC (Casale et al., 2019) | $2.81 \pm 0.03$ | - | 3.7 | 1 | gradient |
| P-DARTS (Chen et al., 2019) | 2.50 | - | 3.4 | 0.3 | gradient |
| CNAS (Lim et al., 2019) | $2.60 \pm 0.06$ | - | 3.7 | 0.3 | gradient |
| ASNGNAS (Akimoto et al., 2019) | - | $2.54 \pm 0.05$ | 3.3 | 0.1 | gradient |
| PC-DARTS (Xu et al., 2020) | $2.57 \pm 0.07$ | - | 3.6 | 0.1 | gradient |
| SDARTS-ADV (Chen & Hsieh, 2020) | - | $2.61 \pm 0.02$ | 3.3 | 1.3 | gradient |
| MergeNAS (Wang et al., 2020b) | - | $2.68 \pm 0.01$ | 2.9 | 0.6 | gradient |
| ISTA-NAS (two stage) (Yang et al., 2020) | $2.54 \pm 0.05$ | - | 3.3 | 0.1 | gradient |
| NASP (Yao et al., 2020) | $2.83 \pm 0.09$ | - | 3.3 | 0.9 | gradient |
| SGAS (Li et al., 2020a) | 2.39 | $2.66 \pm 0.24$ | 3.7 | 0.25 | gradient |
| DrNAS (Chen et al., 2021) | $2.54 \pm 0.03$ | - | 4.0 | 0.4 | gradient |
| DARTS (1st) (Liu et al., 2019) | $3.00 \pm 0.14$ | - | 3.3 | 0.4 | gradient |
| Few Shot + DARTS (1st) | 2.48* | $2.60 \pm 0.10$* | 3.6 | 1.1 | gradient |
| GM + DARTS (1st) | **2.35** | **2.46 ± 0.07** | 3.7 | 1.1 | gradient |
| DARTS (2nd) (Liu et al., 2019) | $2.76 \pm 0.09$ | - | 3.3 | 1.0 | gradient |
| Few Shot + DARTS (2nd) | 2.58* | $2.63 \pm 0.06$* | 3.8 | 2.8 | gradient |
| GM + DARTS (2nd) | **2.40** | **2.49 ± 0.08** | 3.7 | 2.7 | gradient |
| SNAS (moderate) (Xie et al., 2019) | - | $2.85 \pm 0.02$ | 2.8 | 1.5 | gradient |
| Few Shot + SNAS | 2.62 | $2.70 \pm 0.05$ | 2.9 | 1.1 | gradient |
| GM + SNAS | **2.34** | **2.55 ± 0.16** | 3.7 | 1.1 | gradient |

\* Reproduced by running both search and retrain phase under four seeds. Few-Shot NAS adopts a different retrain protocol than the commonly used DARTS protocol; The test accuracy of its released discovered architecture under DARTS' protocol is 2.44%, similar to our reproduced "best" result on DARTS-1st (2.48%).
[†] Obtained on a different search space with PyramidNet (Han et al., 2017) as the backbone.

algorithms [2]. For GM-NAS, we select three edges ($T = 3$) in total based on the edge importance measure introduced in Section 3.4 and split the operations on each selected edge into two groups ($B = 2$). This leads to eight sub-supernets, comparable to the seven supernets used in the Few-Shot NAS baseline. We also restrict our total search cost to match that of Few-Shot NAS for fair comparisons. We refer the readers to Appendix C for more details about the settings.

As shown in Table 4, GM-NAS consistently outperforms Few-Shot NAS on both variants of DARTS and also SNAS. For instance, GM DARTS (1st) achieves a 2.35% test error rate, 0.13% lower than Few-Shot DARTS (1st). In addition, GM-NAS also achieves significantly better average test accuracy (Avg column) than Few-Shot and One-Shot NAS, which shows the robustness of our search algorithm under different random seeds. Notably, the best test error we obtain across different base methods is 2.34% (GM-SNAS), ranking top among prior arts.

## 4.3 MOBILENET SPACE

In addition to the cell-based search spaces, we also evaluate the effectiveness of GM-NAS on the MobileNet Space. Following Few-Shot NAS, we apply GM-NAS to two sampling-based methods - ProxylessNAS (Cai et al., 2019) and OFA (Cai et al., 2020). To match the total number of sub-supernets of GM-NAS with Few-Shot NAS, we select two layers ($T = 2$) to perform the supernet partitioning, and divide the operations into two groups ($B = 2$) for the first edge, and three groups ($B = 3$) on the second edge.

The results are summarized in Table 5. When applied to ProxylessNAS, GM-NAS achieves a Top-1 test error rate of 23.4%, out-performing both Few-Shot and One-Shot versions by 0.7% and 1.5%, respectively. On OFA, we obtain a 19.7% Top-1 test error, surpassing all comparable methods within 600-FLOPs latency. The strong empirical results demonstrate GM-NAS' ability to effectively re-

---

[2]Note that previous methods usually pick the best architecture from up to 10 search runs and report the standard deviation of **only the evaluation phase**. Such reporting schema does not capture the variance of the search algorithm, and thus biases toward highly unstable methods.

Table 5: Comparison with state-of-the-art image classifiers on ImageNet under mobile setting.

| Architecture | Test Error(%) | | Params | Flops | Search Cost | Search |
|---|---|---|---|---|---|---|
| | top-1 | top-5 | (M) | (M) | (GPU days) | Method |
| Inception-v1 (Szegedy et al., 2015) | 30.1 | 10.1 | 6.6 | 1448 | - | manual |
| MobileNet (Howard et al., 2017) | 29.4 | 10.5 | 4.2 | 569 | - | manual |
| ShuffleNet 2× (v1) (Zhang et al., 2018) | 26.4 | 10.2 | $\sim 5$ | 524 | - | manual |
| ShuffleNet 2× (v2) (Ma et al., 2018) | 25.1 | - | $\sim 5$ | 591 | - | manual |
| NASNet-A (Zoph et al., 2018) | 26.0 | 8.4 | 5.3 | 564 | 2000 | RL |
| PNAS (Liu et al., 2018) | 25.8 | 8.1 | 5.1 | 588 | 225 | SMBO |
| AmoebaNet-C (Real et al., 2019) | 24.3 | 7.6 | 6.4 | 570 | 3150 | evolution |
| MnasNet-92 (Tan et al., 2019) | 25.2 | 8.0 | 4.4 | 388 | - | RL |
| GDAS (Dong & Yang, 2019) | 26.0 | 8.5 | 5.3 | 581 | 0.3 | gradient |
| BayesNAS (Zhou et al., 2019) | 26.5 | 8.9 | 3.9 | - | 0.2 | gradient |
| PARSEC (Casale et al., 2019) | 26.0 | 8.4 | 5.6 | - | 1 | gradient |
| P-DARTS (CIFAR-10) (Chen et al., 2019) | 24.4 | 7.4 | 4.9 | 557 | 0.3 | gradient |
| SinglePathNAS (Guo et al., 2019)[†] | 25.3 | - | 3.4 | 328 | 8.3 | evolution |
| EfficientNet-B1 (Tan & Le, 2019)[†] | 20.9 | 5.6 | 7.8 | 700 | - | grid search |
| DSNAS (Hu et al., 2020a)[†] | 25.7 | 8.1 | - | 324 | - | gradient |
| ISTA-NAS (Yang et al., 2020)[†] | 24.0 | 7.1 | 5.7 | 638 | - | gradient |
| PC-DARTS (ImageNet) (Xu et al., 2020)[†] | 24.2 | 7.3 | 5.3 | 597 | 3.8 | gradient |
| BigNASModel-L (Yu et al., 2020a)[†] | 20.5 | - | 6.4 | 586 | - | gradient |
| GAEA + PC-DARTS (Li et al., 2020b)[†] | 24.0 | 7.3 | 5.6 | - | 3.8 | gradient |
| DrNAS (Chen et al., 2021)[†] | 23.7 | 7.1 | 5.7 | 604 | 4.6 | gradient |
| DARTS (2nd) (Liu et al., 2019) | 26.7 | 8.7 | 4.7 | 574 | 1.0 | gradient |
| GM+DARTS (2nd) | **24.5** | 7.3 | 5.1 | 574 | 2.7 | gradient |
| SNAS (mild) (Xie et al., 2019) | 27.3 | 9.2 | 4.3 | 522 | 1.5 | gradient |
| GM+SNAS | **24.6** | 7.4 | 5.2 | 583 | 1.1 | gradient |
| ProxylessNAS (GPU) (Cai et al., 2019)[†] | 24.9 | 7.5 | 7.1 | 465 | 8.3 | gradient |
| Few Shot + ProxylessNAS (Zhao et al., 2021b)[†] | 24.1 | - | 4.9 | 521 | 20.75 (11.7)* | gradient |
| GM + ProxylessNAS[†] | **23.4** | 7.0 | 4.9 | 530 | 24.9 | gradient |
| OFA_Net (Large) (Cai et al., 2020)[†] | 20.3 (20.0)[‡] | 5.1 (5.1) | 9.1 | 595 | 1.7[§] | gradient |
| Few Shot + OFA_Net (Large) (Zhao et al., 2021b)[†] | 20.2 (19.5)[‡] | 5.2 (-) | 9.2 | 600 | 1.7[§] | gradient |
| GM + OFA_Net (Large)[†] | **19.7** | 5.0 | 9.3 | 587 | 1.7 [§] | gradient |

[†] The architecture is discovered on ImageNet directly, otherwise it is discovered on CIFAR-10 (Transfer Setting).

[*] The search cost of Few-Shot ProxylessNAS (20.75) is estimated based on the code we obtained from the author, which is different from the one reported in the original paper (11.7).

[‡] "$x(y)$": $x$ denotes the reproduced results and $y$ is the one reported in the original papers. We refer the reader to Appendix D.3 for discussions on reproducibility.

[§] We follow OFA and Few-Shot OFA paper to report the search cost, which only includes the cost of evolutionary search.

duce the harm of weight-sharing in One-Shot Algorithms on large-scale search spaces and datasets, compared with Few-Shot NAS. We also evaluate the performance of transferring the searched architectures (GM-DARTS and GM-SNAS) in CIFAR-10 to the ImageNet task, which further validates the effectiveness of our GM-NAS methods.

## 5 ABLATION STUDY

In this section, we conduct extensive ablation studies on GM-NAS. Similar to Section 3, we use CIFAR-10 and NASBench-201 space with four operations as it allows us to align the number of supernets with Few-Shot NAS and establish proper comparisons.

### 5.1 THE EFFECT OF DIFFERENT COMPONENTS IN THE PROPOSED METHOD

**Gradient similarity measures** We examine the effect of different similarity measures for computing the gradient matching score on the proposed method. We compare cosine similarity with $l_2$ distance, as well as per-filter cosine similarity (Zhao et al., 2021a) that computes cosine similarity for each convolution filters separately. As shown in Table 6, per-filter cosine similarity performs similarly to cosine

Table 6: Test Accuracy (%) of the derived architectures from GM-NAS with different gradient similarity measures on NASBench-201. Two variants of Cosine similarity perform similarly while $L_2$ distance is not as effective.

| Measure | $L_2$ | Per-Filter-COS | COS |
|---|---|---|---|
| Accuracy | $92.52 \pm 0.93$ | $93.87 \pm 0.11$ | $\mathbf{93.95 \pm 0.08}$ |

similarity while $l_2$ distance is not as effective. We therefore adopt cosine similarity for its simplicity.

**Edge selection using gradient matching score** Instead of randomly selecting edges to partition the supernet on, GM-NAS leverages the cut cost from graph min-cut to decide which edge to partition on. Empirically, we find that this technique reduces the variance and also improves

the search performance compared with random selection. GM-NAS with random edge selection obtains 93.58% mean accuracy with a standard derivation of 0.39 on NASBench-201. In contrast, GM-NAS with the proposed edge selection achieves 93.95% accuracy with only a faction of the variance (0.08).

**Warmup epochs during the Splitting phase**
To obtain accurate gradient information, we warmup the supernet for a few epochs during the supernet partitioning phase. As shown in Figure 2, our proposed method is robust to a wide range of warmup epochs. Note that the variance increases without warmup (warmup epoch = 0) due to noisy gradients at initialization, indicating that proper supernet warmups are necessary for GM-NAS.

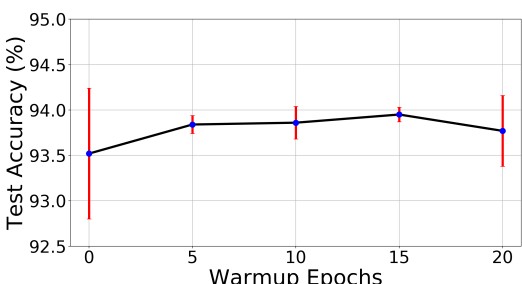

**Restart** After the final split is completed, we re-initialize the weights of sub-supernets before conducting architecture search on them. Intuitively, restarting the sub-supernets eliminate the negative effect of weight-sharing established during the splitting phase. To verify this, we test GM-NAS without restart while keeping

Figure 2: Test Accuracy (%) of the derived architectures from GM-NAS with different supernet warmup epochs during the supernet splitting phase on NASBench-201. The performance of GM-NAS stays stable across different settings.

the number of supernet training epochs identical to the GM-NAS baseline. We obtain a mean accuracy of 91.04% on CIFAR-10, which is 2.91% lower than the original GM-NAS, which shows the necessity of supernet restart.

## 5.2 RANKING CORRELATION

We also evaluate the ranking correlation (measured by Spearman correlation) among top architectures of the proposed method. This is a particularly important measure for effective NAS algorithms as they have to navigate the region of top models to identify the best candidates (Abdelfattah et al., 2021). For Few-Shot NAS, we split two edges, amounting to $4^2 = 16$ sub-supernets. For GM-NAS, due to a smaller branching factor ($B = 2$), we could afford to split four edges while keeping the total number of supernet the same as Few-Shot NAS. We train each sub-supernet using Random Sampling with Parameter Sharing (RSPS) (Li & Talwalkar, 2020), same as the original Few-Shot NAS paper. Table 7 summarizes the results. The proposed method obtains 0.532 Spearman correlation among top 1% architectures, much higher than Few-Shot NAS (0.117). The improved ranking correlation also justifies the superior performance of GM-RSPS over Few-Shot RSPS reported in Section 4.1.

Table 7: Ranking Correlation among top architectures from NASBench-201. The proposed splitting schema leads to significantly better ranking correlation than Few-Shot NAS.

| Method | Branching Factor | #Splits | #Supernets | Spearman Correlation ($\rho$) | | |
|---|---|---|---|---|---|---|
| | | | | Top 0.2% | Top 0.5% | Top 1% |
| Few-Shot | 4 | 2 | 16 | 0.024 | 0.032 | 0.117 |
| GM (ours) | 2 | 4 | | 0.410 | 0.411 | 0.532 |

## 6 CONCLUSION

In this paper, we demonstrate that gradient similarity can effectively measure the harm of weight-sharing among child models. We propose a novel Gradient Matching NAS (GM-NAS) - a generalized supernet splitting schema that utilizes gradient matching score as the splitting criterion and formulates supernet partitioning as a graph clustering problem. Extensive empirical results across multiple prevailing search spaces, datasets, and base methods show that GM-NAS consistently achieves stronger performance than its One-Shot and Few-Shot counterparts, revealing its potential to play an important role in weight-sharing Neural Architecture Search methods.

## ETHICS STATEMENT

We do not aware of any potential ethical concerns regarding our work.

## ACKNOWLEDGEMENT

This work is partially supported by NSF under IIS-2008173, IIS-2048280 and by Army Research Laboratory under agreement number W911NF-20-2-0158.

## REPRODUCIBILITY STATEMENT

We provide a copy of our code in the supplementary material, including both search and retrain phase for our method and the reproduced baseline, to ensure reproducibility on all search spaces. Our experimental setting is stated in Section 4, and hyperparameters are described in the Appendix D. Furthermore, we also include discussions on the reproducibility of relevant baselines on DARTS and MobileNet Space in Appendix D.

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

APPENDIX

## A  PSEUDOCODE FOR GM-NAS

Algorithm 1 summarizes the main pipeline of the proposed Generalized Supernet Splitting with Gradient Matching (GM-NAS). The pseudocode for our partitioning algorithm is provided separately in Algorithm 2 for ease of reference. Note that the main difference between GM-NAS and Few-Shot NAS lies in the **supernet splitting phase**: Compared with Few-Shot NAS' exhaustive partitioning, GM-NAS leverages the well-motivated gradient matching score and the graph min-cut algorithm for making much more informed splitting decisions.

---

**Algorithm 1:** Main Pipeline

**Input:** The set of supernet $\mathcal{A} = \{A\}$, warmup epochs $warm\_epo$ for supernet splitting, number of splits $T$, Branching factor $B$ per split.

    `// supernet splitting phase`
1  $\mathcal{A} = \{A_0\}$;
2  **forall** $t = 1, \cdots, T$ **do**
3      $\mathcal{A}' = \{\}$;
4      **forall** $A \in \mathcal{A}$ **do**
5          Train $A$ for $warm\_epo$ epochs;
6          $\mathcal{A}' \xleftarrow{insert} GM\_SPLIT(A, branch\_factor = B)$;
7      **end**
8      $\mathcal{A} = \mathcal{A}'$;
9  **end**
    `// search phase`
10  reinitialize $w$ in $\mathcal{A}$;
11  perform architecture search on $\mathcal{A}$ using corresponding base methods;

---

**Algorithm 2:** Supernet Splitting with Gradient Matching, $GM\_SPLIT(A, B)$

**Input:** Supernet $A$, Branching factor $B$
1  **forall** *unsplitted edge $e$ on $A$* **do**
2      **forall** *operation $o$ on $e$* **do**
3          temporarily enable $o$ while disabling other operations on $e$;
4          evaluate $A$'s gradient on other shared edges, averaged over $M$ mini-batches;
5      **end**
6      calculate the gradient matching scores based on Eqn.(2);
7      compute $e$'s importance score and operation partition based on Eqn.(3);
8  **end**
9  Select the edge $(e^*)$ with the best importance score to split $A$ on;
10  From $e^*$, partition $A$ into $B$ sub-supernets $\{A_b\}_{b=1\cdots B}$;
11  **return** $\{A_b\}_{b=1\cdots B}$

---

## B  COMPLEMENTARY ANALYSIS

### B.1  WHAT GRADIENT TELLS US ABOUT WHICH OPERATIONS SHOULD (NOT) SHARE WEIGHTS?

On NASBench-201, we find that gradient matching score always decides to assign conv_1x1 and conv_3x3 into one partition (sub-supernet), and Skip and AvgPool_3x3 into another. This behavior is reasonable: High-performing architectures on NASBench-201 usually contains more parametric operations; And weight-sharing between these high performers and the rest of poor architectures could be quite harmful to the former. GM-NAS breaks the weight-sharing between high performers and the rest of the architectures. This has been shown to improve the performance estimation of top models, leading to better search performance (Zhang et al., 2020b).

This is no longer the case on the more complex DARTS Space, where the number of parametric operations is less correlated with the search performance. We observe that three of the four convolution operations are usually assigned to one group, while the other group consists of three non-parametric operations and one convolution. And the specific convolution operation assigned to the second group varies.

For MobileNet space consisting of structured convolution blocks, we did not observe any particular patterns in terms of how GM-NAS decides to partition the supernet. Still, the superior performance of GM-NAS over Few-Shot NAS and One-Shot NAS in Table 5 demonstrates that GM-NAS can make effective partitioning decisions on this search space.

### B.2 More details on the preliminary experiment in Table 2

We provide further details for the experiment in Table 2. Given a target architecture $A$, we construct $A_{sim}$ (and $A_{dissim}$) by changing the operation on a randomly selected edge of $A$ to another operation so that the cosine similarity between gradients computed from $A$ and from $A_{sim}$ ($A_{dissim}$) at shared weight are large (small). Concretely, $cos(\nabla_{w_s}\mathcal{L}(A; w_s), \nabla_{w_s}\mathcal{L}(A_{sim}; w_s)) > 0.7$ for $A_{sim}$ and $cos(\nabla_{w_s}\mathcal{L}(A; w_s), \nabla_{w_s}\mathcal{L}(A_{dissim}; w_s)) < 0.3$ for $A_{dissim}$, where $w_s$ is their shared weight. Note that the gradient similarities are computed by averaging over 100 mini-batches after updating $A$ together with $A_{sim}$ (or $A_{dissim}$) for 2 epochs, in order to obtain a more accurate estimation. We then record and compare the training losses of $A$ when 1) it is updated together with $A_{sim}$ and 2) it is updated together with $A_{dissim}$ for 20 epochs. The above process is repeated for 50 randomly sampled $A$, and the mean training losses and gradient similarities are reported in Table 2. As expected, the training loss of $A$ is much lower in case (1) when $A$ shares weights with $A_{sim}$, since the training dynamics of $A$ and $A_{sim}$ are similar at the shared weight.

### B.3 Ablation studies on graph cut algorithms

For all experiments in the main text, we adopt a simple graph cut algorithm with an explicit constraint for edge/layer partitioning. Notably, the constraint term in Eqn.(3) indicates that the operations are divided into roughly balanced groups. This corresponds to the balancing or normalization factor in more complex graph clustering algorithms such as Ncut (von Luxburg, 2007). We do not use unconstrained graph min-cut algorithms because they tend to degenerate to trivial solutions where the algorithm simply splits one node from the rest (von Luxburg, 2007). Another reason for adopting this constraint in NAS is that we want the sub-supernets to be more balanced, as they will later be trained under identical settings. We also experiment with advanced clustering algorithms such as Ncut, but find that they mostly produce identical cuts to the algorithm described in Eqn.(3). We conjecture that it is because empirically the gradient matching scores often lead to distinguishable clusters, making the result insensitive to the choice of cut algorithms.

## C Supernet Selection

In this section, we address the problem of deriving the final architecture from the set of partitioned sub-supernets, which we term *Supernet Selection*. Recall that Few-Shot NAS (and our method) splits the supernets into $N$ sub-supernets, and then performs architecture search over these sub-supernets by training them independently from scratch before deriving a single final architecture from them.

Few-Shot NAS argues that the architecture selected from the sub-supernet with the lowest validation loss is typically the best among architectures derived from all sub-supernets. However, empirically we find that this is often not the case, especially on irregular search spaces like the cell-based DARTS Space. As shown in Table 8, for Few-Shot NAS, the architecture selected from the sub-supernet with lowest validation loss (or accuracy) does not match the top architecture from all sub-supernets ("max" entry in the Table). This result aligns with previous findings that the supernet's performance is unrelated to the final subnetwork accuracy (Li et al., 2020a). There could be multiple potential reasons for this behavior. For example, the sub-supernet that hosts the top child architecture might also contain many mediocre architectures, so the performance of this sub-supernet as a whole might not necessarily top other sub-supernets. Moreover, after the supernet splitting phase, architecture search is performed on these sub-supernets independently, so their performance are also subject to randomness in the search phase.

One could always evaluate each architecture derived from $N$ sub-supernets and pick the best one out of them, which leads to extra overhead as $N$ increases. To solve this, we propose to adopt Successive Halving (Karnin et al., 2013; Jamieson & Talwalkar, 2016; Li et al., 2019b) to reduce the supernet selection cost. Successive Halving progressively discards half of the poor architectures following a predefined schedule, and stops until only one candidate is left. For DARTS Space, we set the schedule to $(30, 100, 600)$, so that the overhead of retraining $N = 8$ architectures is only $2.5\times$ that of retraining a single architecture ($30 * 8 + 70 * 4 + 500 * 2 = 1520$ epochs). Empirically, we find that varying this schedule has a negligible effect on the recall of successive halving schedules, which aligns with previous discoveries on other NAS search spaces (Wang et al., 2021a). As shown in Table 8, successive halving produces essentially the same results for all methods. We summarize the detailed successive halving procedure in Algorithm. 3.

Note that we only perform successive halving for the DARTS Space; and we apply it to both GM-NAS and Few-Shot NAS for fair comparisons. For MobileNet Space, we follow Few-Shot NAS and use the validation loss for supernet selection, as it produces good enough results and speedup the experiments.

Table 8: Performance comparison among derived child networks using different supernet selection criteria in Few-Shot NAS and GM-NAS

| Method | Supernet selection criterion | Test Error(%) | |
| --- | --- | --- | --- |
| | | Best | Avg |
| Few-Shot DARTS (1st) | validation accuracy | 2.74 | 2.88±0.15 |
| | validation loss | 2.53 | 2.73±0.21 |
| | max | **2.48** | **2.60±0.10** |
| | successive halving | **2.48** | **2.60±0.10** |
| GM-DARTS (1st) | validation accuracy | 2.45 | 2.73±0.17 |
| | validation loss | 2.45 | 2.68±0.17 |
| | max | **2.35** | **2.46±0.07** |
| | successive halving | **2.35** | **2.46±0.07** |
| Few-Shot DARTS (2nd) | validation accuracy | 2.83 | 3.00±0.16 |
| | validation loss | 3.25 | 3.42±0.11 |
| | max | **2.58** | **2.63±0.06** |
| | successive halving | **2.58** | **2.63±0.06** |
| GM-DARTS (2nd) | validation accuracy | 2.57 | 2.68±0.11 |
| | validation loss | 2.59 | 2.68±0.08 |
| | max | **2.40** | **2.49±0.08** |
| | successive halving | **2.40** | **2.49±0.08** |
| Few-Shot SNAS | validation accuracy | 2.49 | 3.15±0.24 |
| | validation loss | 2.88 | 3.32±0.34 |
| | max | **2.62** | **2.70±0.05** |
| | successive halving | **2.62** | **2.70±0.05** |
| GM-SNAS | validation accuracy | 2.49 | 2.70±0.17 |
| | validation loss | 2.49 | 2.72±0.20 |
| | max | **2.34** | **2.55±0.16** |
| | successive halving | **2.34** | **2.55±0.17** |

---

**Algorithm 3:** Supernet Selection via Successive Halving

---

**Input:** A candidate pool $|\mathcal{P}|$ with $N$ child networks derived from $N$ sub-supernets, checkpoint
      schedules $ckpts = \{epo_1, epo_2, ..., epo_T\}$
**Result:** a single network from $P$

1 epoch $epo = 1$;
2 **while** $|\mathcal{P}| > 1$ **do**
3      train each network in $P$ for one epoch
4      $epo \mathrel{+}= 1$;
5      **if** $epo \in ckpts$ **then**
6          calculate the validation accuracy for each network in $P$;
7          discard the bottom half of networks from $P$ based on their validation accuracy;
8 **end**
9 **return** the architecture left in $\mathcal{P}$;

---

# D    IMPLEMENTATION DETAILS

## D.1    NASBENCH-201

For experiments in Section 4.1, we set the warmup epoch to 15 for DARTS and SNAS, and 20 for RSPS as its single path nature requires longer training  (Zhang et al., 2020a). We then perform architecture search on each sub-supernet for 30 epochs (50 for RSPS) following the same protocol of the corresponding base algorithms. The search is conducted on three datasets separately for four random seeds. For all methods, we select the best architecture from the sub-supernets as the final architecture.

## D.2    DARTS SPACE

**Supernet Partition**    During the supernet (with 8 cells) splitting phase, we set the warmup epoch to 2, the number of splits to 3, and the branching factor to 2. After the supernet splitting phase is finished, we conduct architecture search on the generated sub-supernets for 15 epochs. This way, the total number of epochs is $2 * 1 + 2 * 2 + 2 * 4 + 15 * 8 = 134$, similar to Few-Shot NAS (train each of the 7 sub-supernets for 20-25 epochs according to the author).

**Supernet Selection**    After searching architectures on each sub-supernet, we apply the successive halving (Appendix C) to select the top-performed child network derived from eight sub-supernets.

**Retraining settings (architecture evaluation)**    To establish a fair comparison with prior arts, we strictly follow the retrain settings of DARTS (Liu et al., 2019) to evaluate the searched architecture. Concretely, we stack 20 cells to compose the final derived architecture and set the initial channel number as 36. The derived architecture is trained from scratch with a batch size 96 for 600 epochs. We use SGD with an initial learning rate of $0.0025$, a momentum of $0.9$, and a weight decay of $3 \times 10^{-4}$, and a cosine learning rate scheduler. In addition, we also deploy the cutout regularization with length 16, drop-path with probability 0.3, and an auxiliary tower of weight 0.4.

**Reproducing Few-Shot NAS Baseline**    Since Few-Shot NAS does not release its search code for the DARTS Space, we follow the search settings in the original Few-Shot NAS paper to reproduce its result: We randomly select one edge and split the supernet into 7 sub-supernets, and conduct **architecture search** on each sub-supernet for 20 epochs. For a fair comparison with GM-NAS, we also use the successive halving (Appendix C) to select the best architecture from these sub-supernets. The reproduced results we obtain for Few-Shot DARTS is 2.48% (the best column in Table 4). This is worse than the one reported in Few-Shot NAS (2.31%) because Few-Shot NAS uses its own retraining protocol to train and evaluate their search architecture, rather than the DARTS' protocol widely adopted by previous methods (confirmed with the author). Evaluating their released best architecture under DARTS protocol results in 2.44% test error, similar to our reproduced result (2.48%). Nonetheless, GM DARTS achieves 2.35% error rate on DARTS, substantially lower than both of these numbers.

### D.3 MOBILENET

**GM ProxylessNAS / OFA** For both ProxylessNAS and OFA, we set the warmup epoch of GM-NAS to 40. We perform split the supernet twice in total, and set the branching factor to 2 for the first split and 3 for the second split, so that the resulting number of sub-supernets match that of Few-shot NAS. After the supernet partitioning phase, for GM ProxylessNAS, we use the same search settings as ProxylessNAS, except for setting the number of search epochs to 40; For GM OFA, we also follow the same search settings as the original paper.

**Supernet Selection** As mentioned in Appendix C, for Mobilenet Space, we select the sub-supernet with the lowest validation loss and derive the best architecture from this supernet, same as Few-Shot NAS.

**Retraining settings (architecture evaluation)** For ProxylessNAS, we follow the settings of Few-Shot ProxylessNAS (Zhao et al., 2021b; Cai et al., 2019) to train our discovered architecture. Since the architecture evaluation codes for OFA (finetune) and Few-Shot OFA (retrain) are not released, we also use the setting of Few-Shot ProxylessNAS to train and evaluate the derived architecture of OFA (reproduced), Few-Shot OFA (reproduced), and GM-OFA.

**Reproducing OFA and Few-Shot OFA baseline** OFA (Cai et al., 2020) does not release the predictor training and child network finetune code. So we follow the settings of Few-Shot Proxy-lessNAS to reproduce the OFA result. Our reproduced accuracy is 79.7%, comparable to the one (80.0%) reported in the OFA paper.

Few-Shot NAS (Zhao et al., 2021b) also does not release their code and the searched architectures on OFA. We try our best to reproduce their result by communicating with the author, but are still not able to reproduce the reported 80.5% accuracy on ImageNet. One potential reason could be that Few-Shot NAS uses an unreleased powerful teacher network to train the derived architectures. Since we could not obtain this teacher network, we follow the Few-Shot ProxylessNAS setting described above to train and evaluate searched architectures for OFA, Few-Shot OFA, and our methods with the teacher model released in Wang et al. (2020a). The reproduced numbers are reported in Table 5.

## E SEARCHED ARCHITECTURES

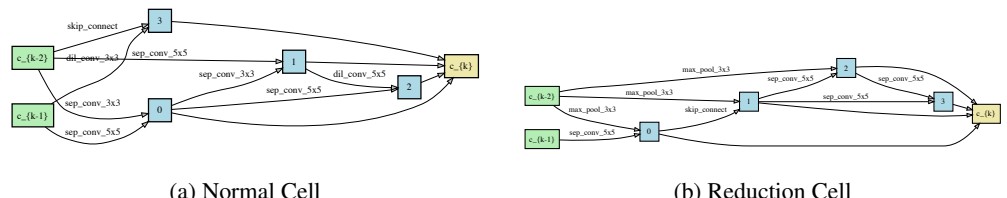

(a) Normal Cell                    (b) Reduction Cell

Figure 3: Normal and Reduction cells discovered by GM-DARTS (1st, seed 0) on CIFAR-10 on DARTS Space

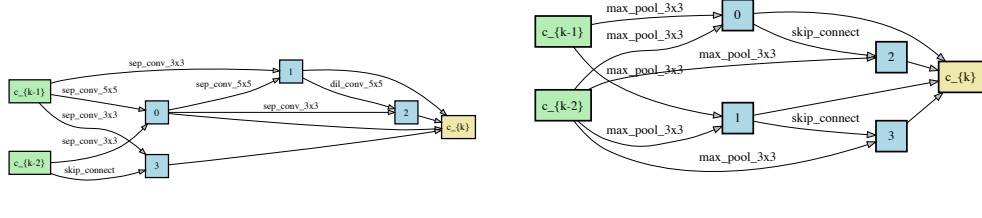

(a) Normal Cell

(b) Reduction Cell

Figure 4: Normal and Reduction cells discovered by GM-DARTS (1st, seed 1) on CIFAR-10 on DARTS Space

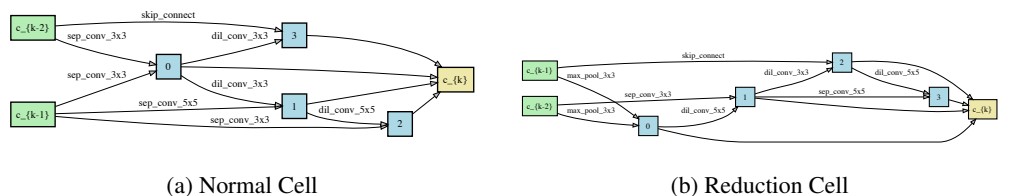

(a) Normal Cell

(b) Reduction Cell

Figure 5: Normal and Reduction cells discovered by GM-DARTS (1st, seed 2) on CIFAR-10 on DARTS Space

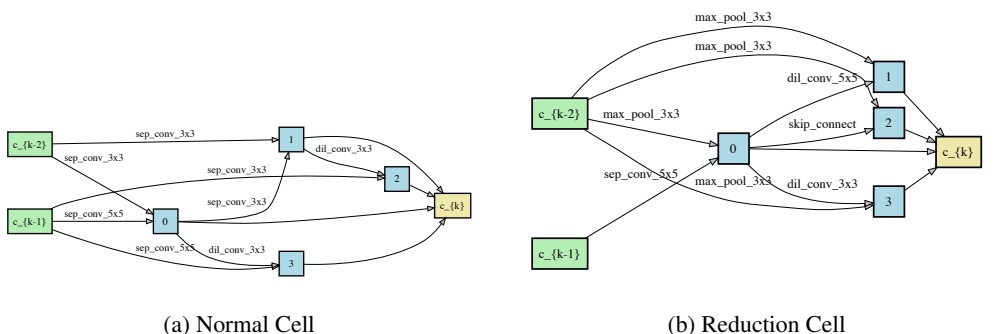

(a) Normal Cell

(b) Reduction Cell

Figure 6: Normal and Reduction cells discovered by GM-DARTS (1st, seed 3) on CIFAR-10 on DARTS Space

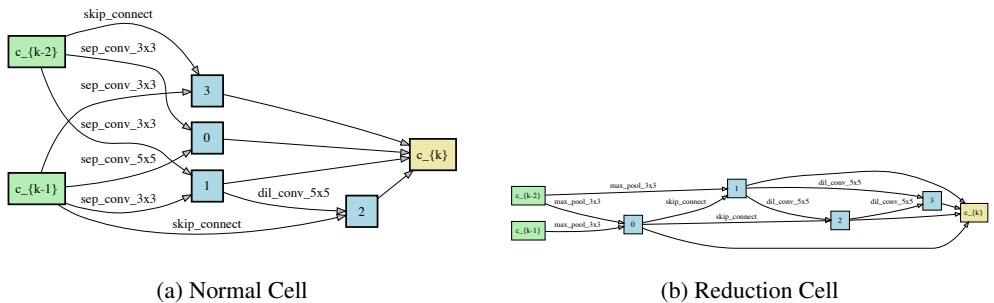

(a) Normal Cell

(b) Reduction Cell

Figure 7: Normal and Reduction cells discovered by GM-DARTS (2nd, seed 0) on CIFAR-10 on DARTS Space

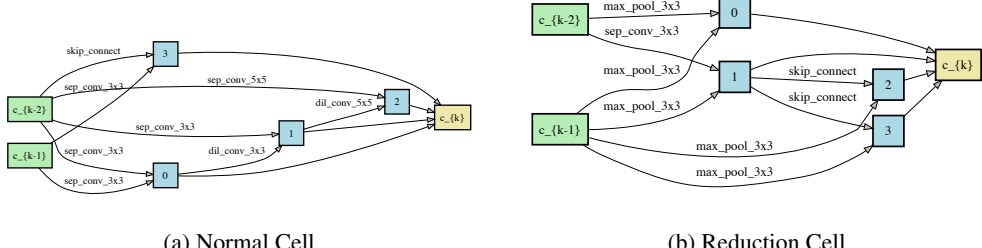

(a) Normal Cell                                    (b) Reduction Cell

Figure 8: Normal and Reduction cells discovered by GM-DARTS (2nd, seed 1) on CIFAR-10 on DARTS Space

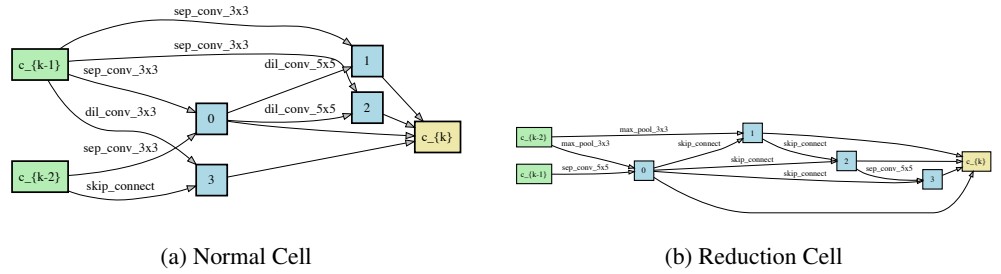

(a) Normal Cell                                    (b) Reduction Cell

Figure 9: Normal and Reduction cells discovered by GM-DARTS (2nd, seed 2) on CIFAR-10 on DARTS Space

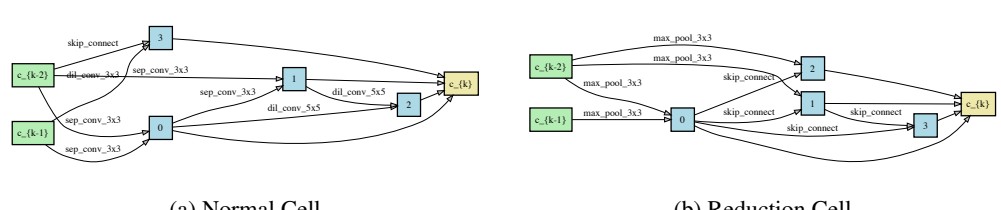

(a) Normal Cell                                    (b) Reduction Cell

Figure 10: Normal and Reduction cells discovered by GM-DARTS (2nd, seed 3) on CIFAR-10 on DARTS Space

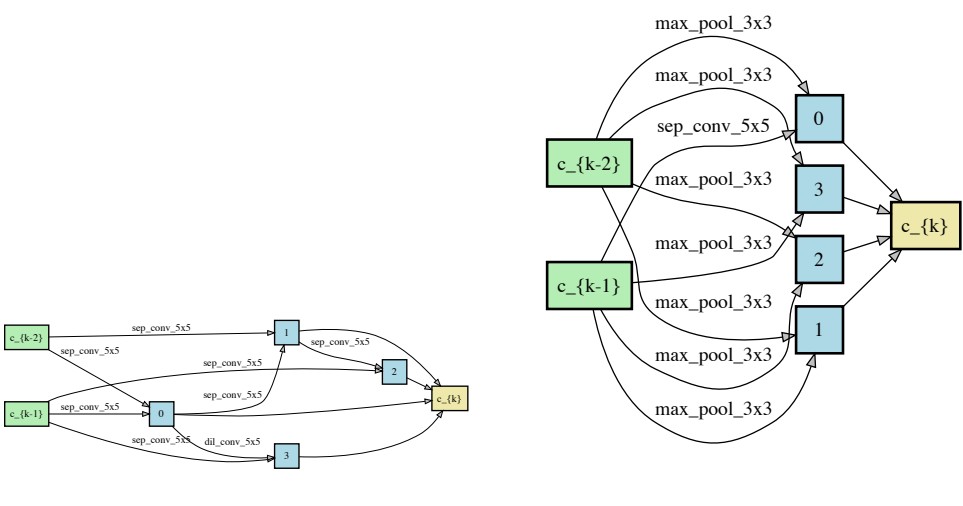

(a) Normal Cell

(b) Reduction Cell

Figure 11: Normal and Reduction cells discovered by GM-SNAS (seed 0) on CIFAR-10 on DARTS Space

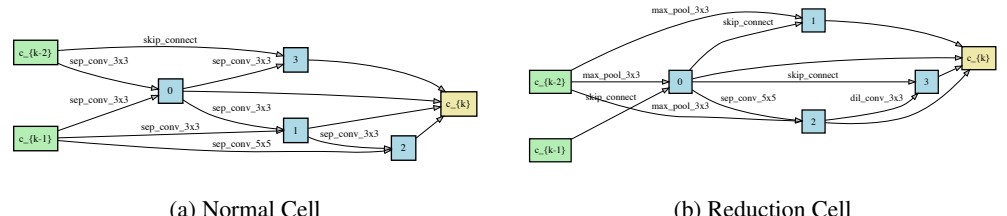

(a) Normal Cell

(b) Reduction Cell

Figure 12: Normal and Reduction cells discovered by GM-SNAS (seed 1) on CIFAR-10 on DARTS Space

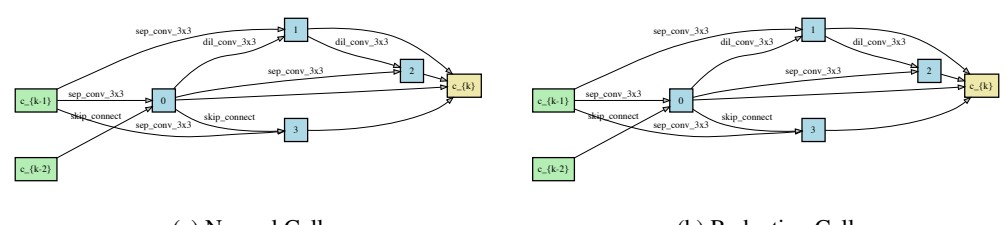

(a) Normal Cell

(b) Reduction Cell

Figure 13: Normal and Reduction cells discovered by GM-SNAS (seed 2) on CIFAR-10 on DARTS Space

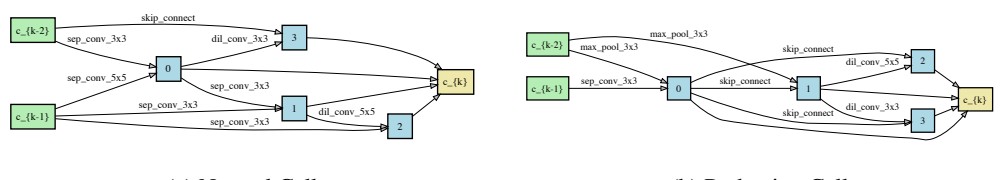

(a) Normal Cell

(b) Reduction Cell

Figure 14: Normal and Reduction cells discovered by GM-SNAS (seed 3) on CIFAR-10 on DARTS Space

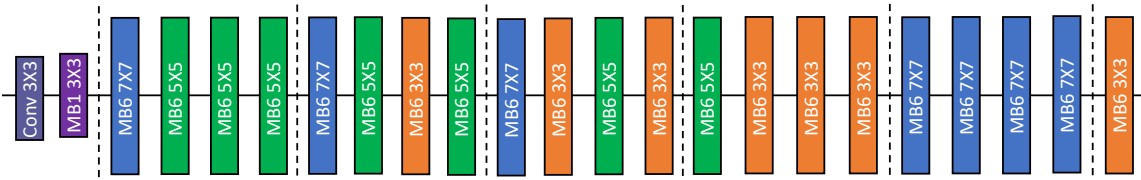

Figure 15: Architecture discovered by GM-ProxylessNAS on ImageNet on MobileNet Space

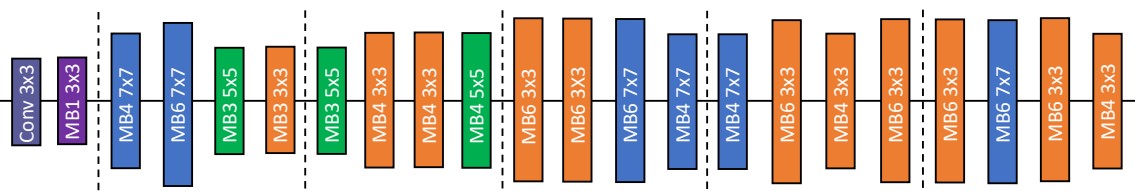

Figure 16: Architecture discovered by GM-OFA (Large) on ImageNet on MobileNet Space

# F    ABLATION STUDY ON THE NUMBER OF SPLITS

We conducted extra ablation study on how the number of splits affect the search outcome of the proposed method using NASBench-201 and CIFAR-10. As showing in Figure 17, GM+DARTS reaches a near oracle search performance with only two splits. Splitting one more time pushes the performance further, but may not worth the extra search cost.

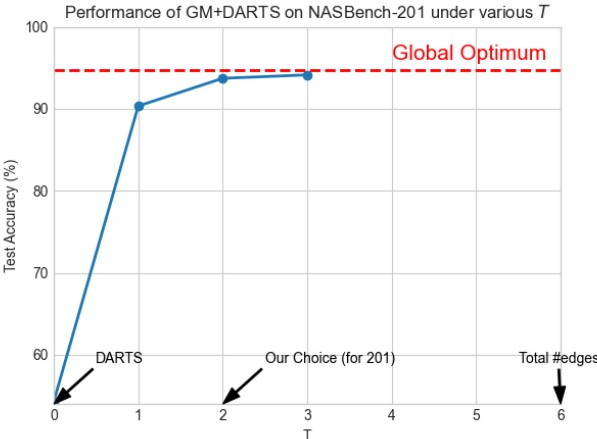

Figure 17: Search performance of GM+DARTS on NASBench-201 and CIFAR-10 under different number of splits $T$. GM-NAS reaches near oracle performance with $T = 2$. Increasing $T$ further brings diminishing return w.r.t. the search cost. All experiments are repeated with four random seeds.

Further ablation study was also conducted in OFA by adding more search budget to GM+OFA on the ImageNet task. We select three layers ($T = 3$) to perform the supernet partitioning, and divide the operations on each selected edge into two groups ($B = 2$). As shown in Tab. 9, this improves GM+OFA further, leading to a test error of 19.4%, which is 0.9% better than OFA and 0.8% better than Few-Shot OFA.

Table 9: Search performance of GM-OFA on ImageNet by adding more search budget

| Method | Top-1 Test Error(%) | Improvement over OFA baseline |
|---|---|---|
| OFA (reproduced) | 20.3 | - |
| Few-Shot OFA (reproduced) | 20.2 | 0.1% |
| GM OFA (ours) | 19.7 | 0.6% |
| GM OFA w/ more search budget (ours) | 19.4 | 0.9% |

