# OpenReview forum: "Generalizing Few-Shot NAS with Gradient Matching"
_ICLR.cc/2022/Conference — ICLR 2022 Poster_

### Official Review · Reviewer_dVgX · 2021-10-20

**Correctness:** 4
**Technical Novelty And Significance:** 3
**Empirical Novelty And Significance:** 3
**Recommendation:** 6
**Confidence:** 4

**Main Review:**

Pros
++ The paper identifies a drawback of the vanilla few-shot NAS, which is the exhaustive partitioning.
++ The paper provides a criterion, ie. graph min-cut on the gradient similarity between operations, to partition the (sub-)supernet.
++ Some toy examples (Table 1 and 2) are provided to support the claim of the paper regarding (1) the necessity of designing a partitioning rule and (2) that the gradient similarity is highly related to the search quality.
++ Extensive experiments show that it is better than the vanilla few-shot NAS. Moreover, it is shown in the ablation study that the ranking correlation in indeed improved by the proposed splitting criterion (Table 7).

Cons
-- The experiments of the proposed GM are mainly designed to match the search budget of few-shot NAS for a fair comparison. But how about the performance of GM when more or less search budge is considered? For example in Table 5, what would happen if less search costs are used when combined with proxylessNAS? Also, what would happen if more search costs are used when combined with OFA_Net?
-- What is the performance of GM+DARTS and GM+SNAS on ImageNet?
-- The improvement over OFA_Net in Table 5 is marginal.

**Summary Of The Paper:**

This paper aims at improving few-shot NAS by proposing a gradient-guided schema to partition the supernet into sub-supernets during the search. The proposed splitting criterion is based on the cosine similarity between the gradients from different child models. Namely, child models that have similar gradients are more likely to be grouped together without splitting, so as to save computational costs and to allow the splitting of more layers than the vanilla few-shot NAS, achieving better performance. Extensive experiments show its superiority over the few-shot NAS.

**Summary Of The Review:**

Overall, it is a paper of good quality. I lean toward accepting the paper because of the intuitive splitting rule proposed in the paper and the supportive results. I will consider raising the rating if the authors can address my above-mentioned questions regarding the experiments.

---

> ### Author Response · Authors · 2021-11-16
> **Response to Reviewer dVgX**
>
> Thank you for your suggestions.
> Up to now, we have finished all but one of the experiments you suggested, and will update the rest as soon as they are ready (The ImageNet training consumes a lot of resources).
> We will also include these extra results in our final draft.
> Please let us know if they address your concerns and whether you have any other questions!
>
> 1. *“The experiments of the proposed GM are mainly designed to match the search budget of few-shot NAS for a fair comparison. But how about the performance of GM when more or less search budge is considered? For example in Table 5, what would happen if less search costs are used when combined with proxylessNAS? Also, what would happen if more search costs are used when combined with OFA-Net? -- What is the performance of GM+DARTS and GM+SNAS on ImageNet? -- The improvement over OFA-Net in Table 5 is marginal.”*
>
>     Thank you for your suggestions:
>
>     a). GM+ProxylessNAS with **reduced search cost** (splitting only one edge) produces 24.1\% Top-1 test error on ImageNet, which is similar to Few-Shot ProxylessNAS.
>     Note that in this case, both methods cut one edge but GM-NAS induces a much lower branching factor ($B=2$) than Few-Shot NAS ($B=6$), which further supports our claim that exhaustively splitting all operations on one edge could be unnecessary.
>     The results of GM+OFA with increased search cost (splitting one more edge) will be updated as soon as they are ready.
>
>     b). We summarize the results for GM+DARTS, GM+SNAS on ImageNet in the Table below. As we can see, GM+DARTS and GM+SNAS achieve 24.5% and 24.6% Top1 test error rates respectively, 2.2\% and 2.7\% better than the corresponding baselines.
>
>     | Base Method 	| Baseline 	| GM-NAS    	| Improvement 	|
>     |-------------	|----------	|-----------	|-------------	|
>     | DARTS       	| 26.7%    	| $\bf 24.5$\% 	| $\bf2.2$\%   	|
>     | SNAS        	| 27.3%    	| $\bf24.6$\%	| $\bf2.7$\%   	|
>
>     c). GM+OFA (19.7\%) improves over Few-Shot+OFA and OFA by 0.5\% and 0.6\% respectively. We would argue that this is quite a significant improvement for ImageNet, especially considering the fact that 1). the FLOPs of our derived architecture is less than Few-Shot+OFA and OFA, and 2). OFA is already the SOTA on MobileNet Space, which makes it much more difficult to improve.
>     Moreover, we expect that adding extra search budgets to GM+OFA would further improve the test error, and we will update the result here once they are finished.

---

> > ### Comment · Reviewer_dVgX · 2021-11-26
> > **Post-rebuttal**
> >
> > I thank the authors for the detailed responses and corresponding experimental results. The author address all my concerns. I would like to increase my rating from 6 to 7.

---

> ### Author Response · Authors · 2021-11-21
> **Final update on the experiment (OFA with more search budget)**
>
> We have finished the GM+OFA with more search budget (split one more edge) you suggested.
> The results are summarized in the Table down below.
> As we can see, adding more search budget improves GM+OFA further, leading to a test error of 19.4\%, which is 0.9\% better than OFA and 0.8% better than Few-Shot OFA. (We refer the reviewer to Table 5 footnote and Appendix D3 for more information on the reproducibility of corresponding baselines.)
>
> Please don't hesitate to let us know if you have any further questions! We sincerely hope the reviewer could take into account the extra experimental results in revising the ratings for the paper.
>
>
> | **Method**                       	| **Test Error (Top-1)** 	| **Improvement over OFA baseline** 	|
> |------------------------------	|--------------------	|-------------------------------	|
> | OFA (reproduced)             	| 20.3%              	| -                             	|
> | Few-Shot OFA (reproduced)    	| 20.2%              	| 0.1%                          	|
> | GM OFA (ours)                      	| 19.7%              	| 0.6%                      	|
> | GM OFA w/ more search budget (ours) 	| 19.4%              	| 0.9%                     	|

---

### Official Review · Reviewer_Y8nY · 2021-11-02

**Correctness:** 3
**Technical Novelty And Significance:** 3
**Empirical Novelty And Significance:** 3
**Recommendation:** 6
**Confidence:** 4

**Main Review:**

This paper is well written and easy to follow. I appreciate the efforts made by this paper on alleviating the weight-sharing problem in one-shot NAS approach by proposing a more reasonable few-shot NAS method from a supernet-splitting perspective. Without many bells and whistles, the proposed method also demonstrates consistent performance gains over its counter-part algorithms.

Though promising, I have three main concerns about this paper:

1, According to section 3.4, greedy algorithm is employed for edge selection, where in each split phase only the edge with maximum sum of GM score is considered to be split. I am doubtful about this specific design because the multi-step split process constitutes a decision chain where local-only criterion usually does not serve for the global purpose. I am willing to see more theoretical (or at least experimental) justification/analysis on this part.

2, All supernet (search space) involved in this paper have orders magnitude more number of edges than three (i.e. the number of edges to be split). Does it suggest the proposed approach can only slightly reduce the weight-sharing extent in the scope of whole supernet? If so, why the searched architectures are extremely close to the optimal one (e.g., Table 3) .

3, There are no ablation studies on the performance changes w.r.t. the different settings of
T
 without considering search cost. It might serve as an important indicator to help understand how much extent the weight-sharing affects the sub-network evaluation accuracy and might also provide some hints for my last question.

Other minors:

1, What is the Random (baseline) method in Table 3. Does it refer to the random partition?

2, Missing T and B settings for NASBENCH-201.

**Summary Of The Paper:**

This paper improves the sub-supernet splitting strategy in few-shot NAS with a gradient matching (GM) score. During supernet optimization, the GM explicitly computes the "agreements" among different operations on an edge in the supernet, which is then treated as the criterion to split those most mismatched operations into different sub-supernets. Although the basic intuition makes some good points and is easy to grasp, I have several concerns on the methodology and the experimental designs of this paper.

**Summary Of The Review:**

This paper presents a new partition strategy for few-shot NAS, which empirically seems to be a more efficient solution to alleviate the weight-sharing problem in one-shot NAS. The intuition/algorithm details are well demonstrated and the experimental evaluation is clearly stated.
Overall I believe the quality of this paper has met the bar of ICLR community, but my concerns about the justifications for its technical designs and some numerical results make me indecisive on posing an acceptance.

I would like to raise my rating if authors properly address my concerns.

---

> ### Author Response · Authors · 2021-11-16
> **Response to Reviewer Y8nY**
>
> Thank you for your feedback.
> We will try our best to address your concerns, and please let us know if you have any further questions!
>
>
> 1. *“According to section 3.4, greedy algorithm is employed for edge selection, ... I am doubtful about this specific design because the multi-step split process constitutes a decision chain where local-only criterion usually does not serve for the global purpose...”*
>
>     We deploy greedy edge selection because after each split, the underlying network changes (i.e. supernet $\rightarrow$ several sub-supernets), and we cannot rely on the old weight for making all edge decisions at once.
>     The newly derived sub-supernets need to be tuned for a few epochs before the next split, so that the weight inherited from the old supernet can adapt to the sub-supernets.
>     Therefore, the true global solution of this multi-split task is brute-force search: applying the above split-tune-split-tune process to all possible sequences of $T$ edges and picking the best one.
>     This global optimal is computationally infeasible to obtain, hence we resort to greedy multi-step split as an approximation.
>
>     Please let us know whether we understand your question accurately and whether the above justifications address your concern.
>
>
> 2. *“All supernet (search space) involved in this paper have orders magnitude more number of edges than three (i.e. the number of edges to be split). Does it suggest the proposed approach can only slightly reduce the weight-sharing extent in the scope of the whole supernet? If so, why the searched architectures are extremely close to the optimal one (e.g., Table 3).”*
>
>     Thank you for your insightful input.
>     We will answer this and the next question together.
>
>     a). First of all, there are actually not that many edges in these search spaces: GM-NAS splits 33.3\%, 21.4\%, and 9.5\% of total edges on 201, DARTS, and MobileNet Space respectively.
>
>     b). We found that the first few splits reduce the adverse effect of weight-sharing a lot, whereas splitting too many edges brings diminishing return w.r.t. the increase of search cost.
>     To show this, we conducted an ablation study on how $T$ affects the performance, following your suggestion in the next question. The resulting figure can be found in Appendix F. As shown in Figure 17, we can reach a near oracle solution with just 2 splits on 201 (6 in total). Splitting 3 edges pushes the performance further but may not be worth the extra search cost.
>
>     c). Another reason why GM-NAS is so effective without splitting all edges should be credited to the descent performance of the base methods (e.g. DARTS, SNAS, etc.).
>     These search algorithms are reasonably effective in the first place, it is just that our paper shows they can be significantly boosted when the level of weight-sharing is reduced.
>     Note that if $T = |\mathcal{E}|$, i.e. split all edges, then GM-NAS reduces to brute-force search, which leads to the global optimal solution.
>     But it would be unnecessary since splitting too many edges lead to diminishing returns, as mentioned in (b).
>
>
> 3. *“There are no ablation studies on the performance changes w.r.t. the different settings of T without considering search cost. It might serve as an important indicator to help understand how much extent the weight-sharing affects the sub-network evaluation accuracy and might also provide some hints for my last question.”*
>
>     (Please see the answer to the above question.)
>
>
> 4. *Minor - "What is the Random (baseline) method in Table 3. Does it refer to the random partition?"*
>
>     “Random (baseline)” refers to "Random Search", provided by NASBench-201.
>     Concretely, it randomly selects a set of architectures, trains them from scratch, and picks the best one based on validation accuracy.
>     Therefore, although the random search baseline on 201 for CIFAR-10 is relatively high, it incurs much more computational cost than one-shot methods and hence does not establish a fair comparison.
>     For weight-sharing NAS, the commonly considered baseline is RSPS (Random Search with Parameter-Sharing) [1], where it trains an one-shot supernet via random sampling to estimate the accuracy of architectures.
>     The performance of RSPS is 87.66\% (Table 3) due to weight-sharing, which is much lower.
>     We will make this point more clear in our final revision.
>
> 5. *Minor - “Missing T and B settings for NASBENCH-201.”*
>
>     The T and B settings for 201 can be found in Section 4.1 Paragraph 1 - “For our method, we split the operations on each edge into two groups (one group with three operations, another group with two operations), and cut two edges in total, amounting to four sub-supernets.”
>     So it would be $T = 2$ and $B = 2$.
>
> [1] Li et al. Random Search and Reproducibility for Neural Architecture Search. (ICML 2019 Workshop)

---

> ### Comment · Reviewer_Y8nY · 2021-11-22
> **Reply to the response**
>
> My concerns are properly addressed. I would like to raise my rating.

---

### Official Review · Reviewer_tbrG · 2021-11-02

**Correctness:** 3
**Technical Novelty And Significance:** 2
**Empirical Novelty And Significance:** 3
**Recommendation:** 6
**Confidence:** 3

**Main Review:**

### Pros:
1. This work answers two questions about how to partion supernets in Few-Shot NAS: 1) Is exhaustive-splitting good? 2) How to partion is better? I think this work has basically answered the above two questions.
2. This work proposes a partioning criteria based on gradient-matching score and formulate the splitting as a graph clustering problem. This partioning method further improves the performance of Few-Shot NAS and  the derived GM-NAS outperforms its Few-Shot and One-Shot counterparts while surpassing previous comparable methods in terms of the accuracy of derived architectures.

### Cons:
1. I don’t quite understand what is the meaning of showing Train Loss in Table 2? Low Train Loss does not mean that the accuracy of the model on the validation set is high. I think you should show the effect of share weight on the model’s performance on the validation set under different Grad Similarity conditions.
2. I don't quite understand why the size of  $\mathcal{U}$ should be restricted in Eqn.(3)?

### Some typos:
1. On the left of Table 2 in Section 3.3, lowercase should be used after the semicolon.


**Summary Of The Paper:**

This work is a further exploration in the direction of Few-Shot NAS. It proposes a method of partioning supernet based on gradient-matching score. Compared with the method based on exhaustive-spltting, the method proposed in this paper can achieve better results. Its contributions are as follows:

1. Point out the problem based on exhaustive-splitting method;
2. A partioning method based on gradient-matching score is proposed, and this method can achieve good results on multiple search spaces and datasets.

**Summary Of The Review:**

Overall, I vote for accepting. First, the overall logic of the paper is relatively clear, basically answering two questions in Pros of **Main Review**. In addition, I think it is reasonable to consider the supernet partition problem from the perspective of gradient, and this work proves the effectiveness of this method through experiments. Hopefully the authors can address my concerns in the rebuttal period.

---

> ### Author Response · Authors · 2021-11-16
> **Response to Reviewer tbrG**
>
> Thank you for your feedback and suggestion.
> We hope our replies address your concerns, and please let us know if you have any further questions!
>
> 1. *“I don’t quite understand what is the meaning of showing Train Loss in Table 2? Low Train Loss does not mean that the accuracy of the model on the validation set is high. I think you should show the effect of share weight on the model’s performance on the validation set under different Grad Similarity conditions.”*
>
>     Thank you for your suggestion.
>     We initially choose to report the training loss as it reflects the convergence of the networks.
>     The behavior is the same on the validation set, and in fact with even larger performance gaps under different gradient similarities:
>
>     | Weight Sharing    	| Grad Similarity 	| Train Loss ($A$)    	| Valid Loss ($A$)  	|
>     |-------------------	|-----------------	|---------------------	|-------------------	|
>     | $(A, A_{sim})$    	| $0.760\pm0.172$ 	| $\bf{0.736\pm0.175}$ 	| $\bf{0.864\pm0.100}$ 	|
>     | $(A, A_{dissim})$ 	| $0.120\pm0.064$ 	| $0.818\pm0.034$     	| $0.994\pm0.027$   	|
>
>
> 2. *“I don't quite understand why the size of $\mathcal{U}$ should be restricted in Eqn.(3)?”*
>
>     The constraint on $\mathcal{U}$’s size indicates that the operations are divided into roughly balanced groups.
>     This corresponds to the balancing or normalization factor in more complex clustering algorithms such as RatioCut and Ncut [1].
>     We did not use unconstrained graph min-cut algorithms because they have been shown to easily degenerate to trivial solutions where the algorithm simply splits one node from the rest [1].
>     Another reason for adopting this design in NAS is that we want the sub-supernets to be more balanced, as they will later be trained under identical settings.
>
> 3. *Minor*
>
>     Thanks for pointing it out! We have fixed the typo.
>
> [1] Luxburg. A Tutorial on Spectral Clustering. (arXiv:0711.0189)

---

### Official Review · Reviewer_x5Hy · 2021-11-07

**Correctness:** 4
**Technical Novelty And Significance:** 2
**Empirical Novelty And Significance:** 3
**Recommendation:** 6
**Confidence:** 3

**Main Review:**

The studied direction is important that we need to endow NAS algorithms quickly adapting to new tasks or data. The work takes a step further based on previous methods. It carefully examines the potential effectiveness improved by grouping operations. The introduced techniques are resonable, like gradients matching, cosine distance, and etc. Overall the paper is very easy to follow and the Figure 2 clearly shows what parts of the problem the method focused on.

In addition, this paper conducted extensive experiments, especially on the comparisons with previous methods, including DARTS, RSPS, SNAS, ProxylessNAS, and OFA and search spaces NASBench-21, DARTS, and MobileNet Space. On three benchmark datasets, it achieved consistent improvements and some of them are significant (Table 3,4,5). Several ablation studies are also conducted, including comparing different similarities measures, edge selection, warmup epoch numbers, and restart.

The reviewer especially likes Appendix B.1 and E where give some instances of the method and the reader can have a certain high-level sense regarding the method. Probably, can also mention some of the contents in the main paper.

Regarding negative points, the reviewer mainly doubts the novelty of the paper since it focused on a small part of the whole problem by utilizing existing tools. Also, the reviewer didn't see many around cost comparisons in the main paper, like time and paranum.

Furthermore, the paper used a very simple algorithm for graph splitting. Probably, more discussion and implementations should be introduced here and add them into the ablation section.


-----------------------------after rebuttal---------------

The reviewer thanks the authors' efforts. Yes, it is figure 1, and the reviewer is sorry for the missing points. But, the reviewer doesn't get convinced by the arguments about the novelty. This work currently studied a relatively small point with existing techniques. It is ok to be accepted, but the reviewer will not fight for it. Also, as the author said, they focused on the splitting criterion, and thus, the reviewer encourages the author to add more comprehensive studies around the graph splitting algorithm.

**Summary Of The Paper:**

This paper proposed a method for few-shot NAS tasks. It found some operations may behave similarly and can be grouped into one and others may not. Those patterns should be considered to save the computation as well as improve accuracy. It proposed a graph construction and spitting pipeline to achieve the goal. Specifically, they used gradient matching and cosine distance to construct the graph among layers. Then, splitting the graph by solving a graph clustering problem.

It conducted experiments on three benchmarks (cifar10, cifar100, and ImageNet) over three search spaces (NASBench-201, DARTS, MobileNet Space) and compared with various baselines. The results show the strength of the proposed method.

**Summary Of The Review:**

Overall, the reviewer thinks the paper is solid while lacking some novelty. The reviewer appreciates the details mentioned in both the main paper and the appendix.

---

> ### Author Response · Authors · 2021-11-16
> **Response to Reviewer x5Hy**
>
> Thank you for your comments.
> We hope our responses address your concerns, and please let us know if you have any further questions!
>
> 1. *“Overall the paper is very easy to follow and the Figure 2 clearly shows what parts of the problem the method focused on.”*
>
>    We wonder if you mean Figure 1?
>
>
> 2. *"Regarding negative points, the reviewer mainly doubts the novelty of the paper since it focused on a small part of the whole problem by utilizing existing tools."*
>
>    To address your concern, we would like the reviewer to consider the following points when assessing the novelty of this paper:
>
>    (a). The Few-Shot NAS task is by itself a novel paradigm, orthogonal to most of previous advancements in One-Shot NAS. **We focus on the most critical part of the Few-Shot NAS paradigm - the splitting criterion**, which was left unaddressed in the original Few-Shot NAS paper. We present not only the first work to address this issue, but also the first work to contribute to Few-Shot NAS paradigm in terms of its algorithmic design.
>
>    (b). The paper also demonstrates an attempt to answer a long overdue question for the widely adopted weight-sharing technique: is there a principled way to decide which networks can share weight? Although gradient is not new, leveraging gradient information for making weight-sharing decisions is novel and non-trivial.
>
>    (c). The simplicity of the resulting algorithm also shows that it requires little human tuning, which is essential to AutoML. As you mentioned, the extensive experimental result shows the strength of the proposed method over comparable One-Shot and Few-Shot methods.
>
>
> 3. *“Also, the reviewer didn't see many around cost comparisons in the main paper, like time and paranum.”*
>
>    We actually included the search time and number of parameters in our main tables following previous works:
>
>    - The search time (measured in GPU days) of corresponding methods can be found in Table 4 and Table 5 under the "search cost" column.
>    As we can see, GM-NAS induces similar search cost as the Few-Shot NAS, while achieving significant performance gains.
>    It is because the graph min-cut with gradient similarity computation can be performed with negligible overhead (roughly under 5 minutes per split).
>
>    - The number of the derived architecture's parameters can be found in Table 4 and Table 5 under the "Params" column.
>    The parameter sizes of GM-NAS's derived architectures are also similar to Few-Shot NAS, and larger than DARTS as DARTS tends to select shallow networks that lack of expressive power [1].
>    Moreover, for ImageNet, the FLOPs is also reported.
>    Following previous works, we restrict the FLOPs of derived architecture on the MobileNet Space to be under 600M for fair comparisons, which is the threshold for Mobile Settings.
>
>
> 4. *“Furthermore, the paper used a very simple algorithm for graph splitting. Probably, more discussion and implementations should be introduced here and add them into the ablation section.”*
>
>    Thank you for your suggestion.
>    We tried more complicated clustering algorithms (e.g. Normalized Cut [2]) but found that they produce mostly the same cut results.
>    We conjecture that it is because empirically the gradient matching scores often lead to distinguishable clusters, making the result insensitive to the choice of cut algorithms.
>    So for simplicity, we adopted the current cut algorithm, which already produces strong results.
>    The implementation is straightforward: we enumerate all possible $\mathcal{U}$ that satisfies the constraint in Eqn. (3), and pick the one that leads to the lowest GM score according to the LHS of Eqn. (3);
>    This is very efficient to do since the number of operations are small, as mentioned in the paragraph above Eqn. (3).
>
>
> [1] Zela et al. Understanding and robustifying differentiable architecture search. (ICLR 2020)\
> [2] Luxburg. A Tutorial on Spectral Clustering. (arXiv:0711.0189)

---

### Author Response · Authors · 2021-11-16
**Revision Summary:**

We thank all reviewers for their positive feedback and constructive suggestions.
For ease of reference, we include the extra experimental results directly inside our responses here, and we will incorporate them into our final draft later.
The only exception is

- A figure for the ablation study on the number of splits $T$, which we include in Appendix F (marked blue).

We add it to the end of the Appendix to avoid any changes to the original section/figure numbering.
We sincerely hope that our responses answer the reviewers' questions, and will address any further concerns you might have.

---

### Author Response · Authors · 2021-11-19
**Follow-up Discussions**

Dear reviewers,

Since the first discussion period will end soon, we wonder if our responses address your questions adequately? If you have any updates or follow-up questions on the paper, please feel free to let us know so that we could reserve enough time to respond to any further comments you might have.

Thanks again for your detailed reviews and constructive comments!

Best,\
Paper2125 Authors

---

### Author Response · Authors · 2021-11-28
**Final discussion follow-up**

Dear reviewers,

We thank all reviewers for their participation during the discussion period. And we are delighted that your concerns and questions are properly addressed and that your ratings are further improved. We will incorporate the valuable discussions here in our final draft. Moreover, we will be here to answer any further questions you might have during the remaining time.

Best Regards,\
Paper2125 Authors

---

### Decision · Program_Chairs · 2022-01-20

**Decision:**

Accept (Poster)

**Comment:**

All reviewers give acceptance scores.
One reviewer also commented that they would like to increase their score from 6 to 7 (which isn't possible in the system).
I encourage the authors to add the substantial new results generated during the rebuttal into the paper.